# Morphological and Chemical Factors Related to Western Flower Thrips Resistance in the Ornamental Gladiolus

**DOI:** 10.3390/plants10071384

**Published:** 2021-07-06

**Authors:** Dinar S. C. Wahyuni, Young Hae Choi, Kirsten A. Leiss, Peter G. L. Klinkhamer

**Affiliations:** 1Plant Science and Natural Products, Institute of Biology (IBL), Leiden University, Sylviusweg 72, 2333BE Leiden, The Netherlands; dinarsari_cw@staff.uns.ac.id; 2Pharmacy Department, Faculty Mathematics and Natural Sciences, Universitas Sebelas Maret, Jl. Ir. Sutami 36A, Surakarta 57126, Indonesia; 3Natural Products Laboratory, Institute of Biology, Leiden University, Sylviusweg 72, 2333BE Leiden, The Netherlands; y.choi@chem.leidenuniv.nl; 4College of Pharmacy, Kyung Hee University, Seoul 02447, Korea; 5Business Unit Horticulture, Wageningen University and Research Center, Postbus 20, 2665ZG Bleiswijk, The Netherlands; kirsten.leiss@wur.nl

**Keywords:** Gladiolus, *Frankliniella occidentalis*, host plant resistance, morphological markers, mesophyll, epidermis, papillae, eco-metabolomics

## Abstract

Understanding the mechanisms involved in host plant resistance opens the way for improved resistance breeding programs by using the traits involved as markers. Pest management is a major problem in cultivation of ornamentals. Gladiolus (*Gladiolus hybridus* L.) is an economically important ornamental in the Netherlands. Gladiolus is especially sensitive to attack by western flower thrips (*Frankliniella occidentalis* (Pergande) (Thysanoptera:Thripidae)). The objective of this study was, therefore, to investigate morphological and chemical markers for resistance breeding to western flower thrips in Gladiolus varieties. We measured thrips damage of 14 Gladiolus varieties in a whole-plant thrips bioassay and related this to morphological traits with a focus on papillae density. Moreover, we studied chemical host plant resistance to using an eco-metabolomic approach comparing the ^1^H NMR profiles of thrips resistant and susceptible varieties representing a broad range of papillae densities. Thrips damage varied strongly among varieties: the most susceptible variety showed 130 times more damage than the most resistant one. Varieties with low thrips damage had shorter mesophylls and epidermal cells, as well as a higher density of epicuticular papillae. All three traits related to thrips damage were highly correlated with each other. We observed a number of metabolites related to resistance against thrips: two unidentified triterpenoid saponins and the amino acids alanine and threonine. All these compounds were highly correlated amongst each other as well as to the density of papillae. These correlations suggest that papillae are involved in resistance to thrips by producing and/or storing compounds causing thrips resistance. Although it is not possible to distinguish the individual effects of morphological and chemical traits statistically, our results show that papillae density is an easy marker in Gladiolus-breeding programs targeted at increased resistance to thrips.

## 1. Introduction

Sustainable growth and development require minimizing the natural resources and toxic materials used, and the waste and pollutants generated, throughout the entire production and consumption process. This also applies to the production of food and ornamentals, the sustainable production of which requires minimizing the use of pesticides. Breeding for resistance becomes more and more important in this respect. Nowadays, we can apply molecular tools, such as gene expression and the use of mutants, to many species to discover mechanisms of host plant resistance. Such methods and techniques have become increasingly cheaper and, at some point, will become available for all crops. However, the need to reduce pesticides is extremely urgent as was recently again signaled by reports on the alarming decline in insect species [1]. For some crops, especially polyploids and crops with large genomes, feasible, required molecular tools will most likely not become available soon. So, the market still calls for fast and cheap alternatives such as morphological or chemical markers.

Plant defense against insect herbivores comprises morphological traits, such as spines, trichomes, and papillae, as well as chemical traits. Trichomes are epidermal hairs protruding from the surface of leaves and stems [2], impeding movement or trapping herbivorous insects, resulting in their death [3,4,5]. Papillae are protuberances of solid cell wall thickening [3]. They are known to protect cells from pathogen attack due to physical barriers [4,5]. Prum et al. [6] showed that papillae made it more difficult for beetles to attach to the leaves, suggesting that papillae also play a role in defense against insect herbivores. Despite this potential role, papillae have not been studied in great detail yet in relation to insect plant defense.

Besides physical barriers, both trichomes and papillae can produce secondary metabolites that deter or are toxic to herbivores. Among those secondary metabolites are glucosinolates, alkaloids, phenolics, phenylpropanoids, polyketides, and terpenoids [7]. For instance, acylsugars [8] and phenols [9] are produced in trichomes of *Solanum pennellii*, while phenolics and alkaloids were detected in trichomes of *Withania somnifera* L. [10]. Papillae store plant defense compounds such as 2-acetyl-1-pyrroline in rice [11] and cardosin A in *Cynara cardunculus* L. [12].

Because trichomes and papillae are part of the plant’s defense system, they may play an important role in breeding programs aimed at increasing natural host plant resistance. Host plant resistance becomes increasingly important in integrated pest management programs directed at agricultural and horticultural key pests such as western flower thrips (*Frankliniella occidentalis*). This invasive pest is highly polyphagous and attacks fruits, vegetables, and ornamentals [13,14]. It has been recorded feeding on over 250 crop species from over 60 families (15), causing losses of millions of euros worldwide [14,15,16]. Thrips have piercing–sucking mouthparts allowing them to suck up a whole cell’s contents, leaving an empty cell filled with air, causing a characteristic silver leaf scar, the so-called silver damage [17]. Adult thrips and larvae feed in the same manner, both contributing to the damage [14]. Females feed more intensively than males, which is attributed to their lower mobility and high consumption rates needed for egg production [18]. Feeding on actively growing tissues leads to stunting and distorted plant growth with eventual yield loss [19]. In addition, feeding injury causes a reduction in the aesthetic value and storage quality of the produce [20]. In addition, thrips is an important vector of viral diseases [21]. 

Host plant resistance to western flower thrips is a promising approach hampering preference, reproduction, feeding, and/or transmission of virus [22]. It is mainly chemically based in a number of plants species, as shown by Leiss et al. [23] applying an eco-metabolomic approach. They compared the ^1^H nuclear magnetic resonance (NMR) spectroscopy profiles of thrips-resistant and thrips-susceptible plants to identify metabolites related to constitutive thrips resistance in wild *Jacobaea* species [24], in the ornamental chrysanthemum [25], and in the vegetables tomato [9,26] and carrot [27], which showed alkaloids, acylsugars, flavonoids, amino acids, and phenylpropanoids, respectively. In addition, eco-metabolomics has been applied to determine metabolites related to thrips resistance in sweet pepper [28,29] and onion [30], which showed triterpenoid derivatives. 

*Gladiolus*, a genus of perennial bulbs, belongs to the Iridaceae family. It is a popular decorative plant in summer and thus constitutes an economically important flower crop in the Netherlands. *Gladiolus* comprises 5% of the total Dutch flower production, constituting 21,000 ha of the production area and amounting to $756 million production in value [31]. Western flower thrips is a major problem in the cultivation of Gladiolus, affecting corms, leaves, buds, and flowers. Thrips damage results in small corms, which may not germinate; problems of flower formation and opening; and an undesirable silvery, shiny damage on leaves and flowers [19].

The aim of this study was to identify morphological and chemical markers for resistance to western flower thrips, looking at feeding damage, that can be used in breeding programs of Gladiolus. In particular, we wanted to address the following questions: (1) Does thrips damage vary among Gladiolus varieties? (2) Is thrips damage related to morphological traits? (3) Is thrips resistance based on chemical traits, and if so, which compounds are involved? (4) Are the morphological and chemical traits mutually correlated? (5) Which (combinations of) traits provide the best marker for thrips resistance in breeding programs? We focused on the following morphological traits: plant dry mass, leaf length, size of epidermal cells, size of mesophylls, and density of papillae at the leaf surface. After we established that thrips resistance is related to several morphological traits, we continued using leaf extracts to show that resistance was at least partly based on plant chemistry. We then compared the ^1^H NMR profiles and thrips resistance of gladiolus varieties representing a broad range of papillae densities to identify potential metabolites related to resistance.

## 2. Results

### 2.1. Differences in Resistance to Thrips

Thrips silver damage in the whole-plant bioassay differed significantly among varieties (F = 11.445, df = 13, *p* = 0.000). Charming Beauty and Charming, as the most susceptible varieties, showed significantly more damage compared with all other varieties, while Robinetta and Alba showed almost no damage at all (Figure 1). Charming displaying the highest amount of damage (mean 3159.3 ± 434.8 mm^2^), showed 130 times more damage than Robinetta (mean 23.8 ± 8.9 mm^2^).

#### 2.1.1. Differences in Morphological Traits

Leaf length (F = 15.522, df = 13, *p* = 0.000) (Figure 2A), the length of epidermal cells (F = 125.459, df = 13, *p* = 0.000) (Figure 3A) and mesophylls (F = 90.136, df = 13, *p* = 0.000) (Figure 3B), and the density of epicuticular papillae (F = 29.363, df = 13, *p* = 0.000) all differed significantly between varieties. These morphological characteristics were mutually highly correlated. This can be explained by the fact that as a rule, each epidermal cell produces one papilla (Figure 4E,F). The different leaf cell lengths as well as papillae density of the thrips-susceptible variety Charming Beauty compared with the thrips-resistant variety Robinetta are depicted as microscopy images in Figure 4A–F. In general, susceptible varieties had longer leaves and cells and lower densities of papillae.

In addition, the dry mass of varieties differed significantly (F = 70.531, df = 13, *p* = 0.000), with large size varieties yielding more than double the dry mass compared with the small ones (Figure 2B). Dry mass was not significantly correlated with the other morphological characteristics.

#### 2.1.2. The Relationship between Thrips Damage and Morphological Characteristics

Silver damage was significantly positive correlated with the length of the epidermal cells (*r* = 0.596, *N* = 14, *p* = 0.024) (Figure 5A) and mesophylls (*r* = 0.603, *N* = 14, *p* = 0.022) (Figure 5B), while it was significantly negatively correlated with the density of papillae (*r* = -0.628, *N* = 14, *p* = 0.016) (Figure 5C). Silver damage did not correlate with leaf length (*r* = 0.320, *N* = 14, *p* = 0.264) or plant dry mass (*r* = −0.222, *N* = 14, *p* = 0.445).

### 2.2. Chemical Study

#### 2.2.1. Differences in the Effect of Six Leaf Extracts on Thrips Mortality

To show that thrips resistance in Gladiolus is at least partly based on chemical characteristics, we studied the effects of leaf extracts of six varieties in artificial diets on thrips mortality. The extracts of the following four varieties with low thrips damage in the previously described whole-plant non-choice bioassay lead to significantly higher thrips mortality compared to the negative control (Figure 6A): Alba (χ^2^= 7.59, d.f. = 1, *p* = 0.005) Nymph (χ^2^= 10.26, d.f. = 1, *p* = 0.001), Elvira (χ^2^= 13.17, d.f. = 1, *p* = 0.0003), and Robinetta, (χ^2^= 8.89, d.f. = 1, *p* = 0.002). The extracts of two varieties with high silver damage, Charming Beauty and Charming, showed a thrips mortality comparable to the negative control. 

Thrips mortality in this in vitro bioassay was negatively correlated with thrips silver damage in the whole-plant non-choice bioassay (*r* = −0.788, *N* = 6, *p* = 0.031) (Figure 6B). This result implied that chemical compounds play a role in plants’ resistance to thrips. We, therefore, continued with the chemical profiling of all varieties.

#### 2.2.2. Metabolic Profiling

In addition to variation in thrips resistance, the Gladiolus varieties differed in their metabolomic profiles. PCA, which is an unsupervised method, did not give a separation of metabolic profiles based on thrips resistance of the varieties (Appendix A). The supervised orthogonal partial least squares (OPLS)–discriminant analysis (DA), in contrast to PCA, takes into account, in addition to the metabolomic matrix, the resistance matrix. We, therefore, separated varieties into resistant (0.03–0.8% damage), partially resistant (4–20% damage), and susceptible (>65% damage) varieties. The loading plot identified candidate signals related to thrips resistance, as shown in the positive value, and candidate signals related to susceptibility, as shown in the negative value (Figure 7). Validation of OPLS-DA by permutation tests resulted in variance *R*^2^ = 0.93 and predictive ability *Q*^2^ = 0.88. *Q*^2^ values greater than 0.5 are generally accepted as good [32].

#### 2.2.3. Signals Related to Resistance

The signals related to thrips resistance were observed in the region of 0.80–1.92 ppm (Figure 7). Compounds showing signals in this region are terpenoids, saponins, and amino acids. Signal A (δ 1.28) and signal B (δ 0.90) were associated with thrips-resistant varieties and belonged to the triterpenoid saponins. However, these remained unidentified due to overlapping signals in ^1^H NMR spectra. We further identified signals related to several amino acids: valine (δ 1.06), alanine (δ 1.48), and threonine (δ 1.32). In addition, the signal of sucrose (δ 5.40) was also related to thrips resistance.

The relative concentrations of alanine, valine, and threonine differed significantly among varieties (F = 21.754, df = 13, *p* = 0.000; F = 75.824, df = 13, *p* = 0.000; and F = 31.460, df = 13, *p* = 0.000). The relative concentrations of alanine, valine, and threonine were three to four times higher in resistant varieties (Table 1). The relative concentrations of alanine and threonine were negatively correlated to thrips silver damage (*r* = −0.612, *N* = 14, *p* = 0.020 and *r* = −0.634, *N* = 14, *p* = 0.015, respectively), while the relative concentration of valine was not significantly correlated to thrips silver damage (*r* = −0.100, *N* = 14, *p* = 0.734).

The relative concentrations of signals A and B were significantly different among varieties (*F* = 52.216, df = 13, *p* = 0.000 and *F* = 44.563, df = 13, *p* = 0.000). Signals A (*r* = −0.505, *N* = 14, *p* = 0.065) and B (*r* = −0.557, *N* = 14, *p* = 0.038) (Table 2) were negatively correlated with silver damage, although for signal A, this was only marginally significant (Table 2). The relative concentration of sucrose differed among varieties (*F* = 14.367, df = 13, *p* = 0.000). Although the concentration of sucrose was about 1.15 times higher in resistant varieties than in susceptible varieties, we did not detect a significant correlation between silver damage and sucrose concentration (*r* = 0.083, *N* = 14, *p* = 0.779) (Appendix A). In conclusion, three of the compounds related to resistance were confirmed to be important in subsequent analyses of relative concentrations of the single compounds in univariate correlations. These compounds were the amino acids alanine and threonine and the compound related to signal B (δ 0.90).

#### 2.2.4. Signals Related to Susceptibility

Glucose was related to susceptibility (Figure 7) and differed significantly (F = 8.352, df = 13, *p* < 0.000 and F = 8.234, df = 13, *p* < 0.000) among varieties (Appendix A). These signals were, however, not correlated to thrips silver damage (*r* = 0.234, N = 14, *p* = 0.420 and *r* = 0.265, *N* = 14, *p* = 0.360) when tested as a single factor (Table 2). Epicatechin, epigallocatechin, and gallic acid were also related to susceptibility. These signals were not detectable in all varieties. The relative concentrations of epicatechin, epigallocatechin, and gallic acid differed significantly among varieties (X^2^ = 52.132, df = 13, *p* = 0.000; −X^2^ = 49.133, df = 13, *p* = 0.000; and X^2^ = 48.397, df = 13, *p* = 0.000, respectively). Epicatechin was marginally significantly related to thrips resistance (ρ = 0.541, *N* = 14, *p* = 0.046); it was present, however, in only three varieties. Epigallocatechin and gallic acid were not related to thrips silver damage when tested as a single factor (ρ = 0.404, *N* = 14, *p* = 0.152 and ρ = 0.313, *N* = 14, *p* = 0.276, respectively) (Table 2). In conclusion, none of the signals related to susceptibility was clearly confirmed to be of significance when tested with univariate correlation tests. 

#### 2.2.5. Correlations between Chemical and Morphological Characteristics Related to Thrips Resistance

All of the metabolites related to resistance were strongly correlated with each other. In addition, they were strongly correlated with the density of papillae. Remarkably, none of the compounds that were associated with susceptibility were correlated with the density of papillae. However, the high correlation amongst compounds prevents the use of multiple regression. The highest-explained variance for thrips resistance was explained by the density of papillae, alanine, threonine, and the compound related to signal B (δ 0.90).

## 3. Discussion

*Gladiolus* varieties showed a broad range of variation in thrips, resistance as demonstrated by the a more than 130-fold difference in silver damage between the most resistant and the most susceptible variety. Such a large variation is not uncommon for ornamentals. Chrysanthemum varieties also exhibit around 100-fold variation in thrips damage [33,34]. Gaum et al. [35] observed that variation in thrips resistance was six times lower in resistant varieties compared with susceptible ones in a study on 25 rose varieties.

The density of papillae was negatively correlated with thrips damage, while the length of mesophylls and epidermal cells was positively correlated with thrips damage. As a rule, an epidermal cell produces one papilla. Thus, varieties with shorter leaf cells have a higher density of epicuticular papillae. Statistically, it is not possible to distinguish between the effects of cell length and density of papillae on silver damage. Papillae may inhibit the movement of thrips or hinder penetration of the epidermis while feeding. However, Prüm et al. (2013) reported that papillae may slightly enhance adhesion to leaves in the Colorado beetle. In line with our study, Scott Brown and Simmonds [36], who studied the effects of leaf morphology on *Heliothrips haemorrhoidalis*, reported that this thrips has a preference for leaves with smooth surfaces, while trichomes and leaf surface wax structures inhibit thrips. Trichomes were also implicated to be related to thrips resistance in tomato [37] and chili peppers [38].

Besides forming a physical barrier, papillae may store plant secondary compounds. The epidermal papillae of *Pandanus amaryllifolius* Roxb. are the storage site of the basmati rice aroma compound, 2-acetyl-1-pyrroline [11]. Cardinosin A, an aspartic proteinase, suggested to be involved in plant defense against pathogens, is stored in the stigmatic papillae of *Cynara cardunculus* L. [12]. Similarly, Gladiolus varieties with higher densities of papillae may contain higher amounts of defense compounds. The density of papillae explained 39% of the variation in silver damage. The correlation between papillae density and thrips damage shows that the density of papillae sets an upper limit to silver damage. However, other factors may be involved as well, as could be seen from the two varieties with low silver damage but relatively low density of papillae. This leads to false negatives when this morphological marker is the only marker used in breeding programs for thrips resistance in Gladiolus. In addition to papillae, chemical traits are likely candidates to be involved in thrips resistance.

We showed that variation in the plant’s metabolome caused variation in thrips mortality in in vitro bioassays. This variation was highly correlated with thrips damage in the whole-plant bioassays. We identified two amino acids and two triterpenoid saponins that were associated with thrips resistance by correlating their relative leaf concentrations with thrips resistance of varieties differing in papillae density. All the compounds that were correlated with resistance were highly correlated amongst each other as well as with papillae density. Remarkably, no compound was clearly related to thrips susceptibility in univariate analyses. One explanation for the combination of these results could be that papillae are involved in resistance to thrips by producing or storing the compounds causing resistance. If this, indeed, is the case, it would also suggest that the physical effect of papillae on thrips resistance is relatively small because, e.g., threonine explains slightly more of the variation in thrips resistance than the density of papillae. Threonine explained 40% of the variation in silver damage. However, the strong correlation among factors identified as being associated with thrips resistance makes it difficult to separate their effects from each other. Likewise, due the correlations between compounds as well as metabolites and density of papillae, no particular single compound can be pinpointed to be related to thrips resistance. Further study on how papillae deter thrips by producing or storing metabolites related to resistance is necessary. Papillae as storage sites of plant defense secondary compounds have been reported in rice [11] and cardoon [12]. In addition to papillae, it is known that compounds produced in leaf trichomes, such as acylsugars, contribute to thrips resistance in wild tomato [9]. Tomato lines bred with increased amounts of acylsucrose show decreased oviposition by western flower thrips and suppressed inoculation with tomato spotted wilt virus [39]. Although tomato trichomes have been reported to be rich in terpenes, containing up to eight different monoterpenes, none of the trichome exudates are related to thrips resistance [26]. It should be pointed out, however, that from our correlative studies, we cannot exclude an alternative hypothesis. For instance, a low density of papillae could result in higher feeding levels, which, in turn, leads to a rapid induction of defense metabolites. If this hypothesis were true, this could also cause a correlation between defense metabolites and papillae density. A logical follow-up of our study would be to analyze the metabolites in the papillae themselves rather than in the whole leaf. This can be further examined by histochemical studies or by analyzing the expression of genes that encode the committed steps in the synthesis of triterpenoid saponins [40]. This offers a promise for further research on the mechanisms involved in resistance. In addition, it should be noted that the factors we identified only explain 40% of the variation in thrips resistance. Most likely, other factors play an additional role in the defense of Gladiolus against thrips. We used NMR analyses. While this has the advantage of an unbiased approach, the sensitivity is relatively low, which may have resulted in important metabolites with respect to defense not being detected.

Both saponins and the amino acids alanine and threonine have been mentioned in the literature in relation to resistance to insect herbivores. The concentration of alanine was higher in a peach variety resistant to the Mediterranean fruit fly compared with a susceptible variety, while for threonine, such a difference was not detected [41]. In addition, Leiss et al. [27] reported that alanine and threonine occur in higher concentration in the leaves of thrips-resistant carrots, leading to higher thrips mortality. In contrast, Dillon and Kumar [42] reported that the concentration of threonine is significantly higher in *Sorghum bicolor* seedlings resistant to the stem borer *Chilo partellus* than in the seedlings of a susceptible variety, while alanine concentrations do not significantly differ. These results confirm the notion that these amino acids may be involved in thrips resistance. However, the correlation we determined for the two amino acids alanine and threonine does not necessarily mean that these compounds confer resistance to thrips. Biosynthetically, these amino acids share the precursor acetyl CoA with the pathway of triterpenoid saponins CoA [43]. It is likely that that these amino acids are associated with a pool of metabolites that support the synthesis of compounds such as triterpenoid saponins, which may act as resistance metabolites. More detailed metabolomic studies, including fluxomics, can shed more light on this.

Saponins are well known to infer resistance to plant herbivores. Saponins were shown to be important defensive chemical in *Aesculus pavia* against the leafminer *Cameraria ohridella* [44]. This leafminer causes heavy damage to the white-flowering horse chestnut in Europe. Among the Aesculus genus, *A. pavia* L., an HBT genotype, characterized by red flowers, showed an atypical resistance toward this pest. This resistance appeared to be based on exogenous saponins that were translocated from roots/stem to the leaf tissues. Saponins have been reported to mediate the resistance in *Barbarea vulgaris* and counter adaptations in the flea beetle *Phyllotreta nemorum* [45,46]. Higher concentrations of triterpenoid saponins in *B. vulgaris* increased resistance to the diamondback moth *Plutella xylostella* as well as western flower thrips, resulting in significantly fewer adults and larvae [47]. Saponins from resistant varieties of garden pea inhibited development of the Azuki bean beetle *Callosobruchus chinensis*, whereas saponin extracts from non-resistant legumes did not [48]. The mechanism through which saponins contribute to resistance are largely unknown. Ishaaya [49] suggested that they slow down the passage of food through the gut, whereas Shaney et al. [50] suggested that saponins block the uptake of sterols, an essential compound that insects cannot synthesize but have to take up through feeding. De Geijter et al. [51] reviewed the effects of saponins on insect herbivores and concluded, “These interesting plant compounds offer new strategies to protect crops in modern agriculture and horticulture with integrated pest management (IPM) programs against pest insects, either by spraying or by selecting high-saponin varieties of commercial crops.”

Our study indicates that both some chemical compounds and papillae density show a strong negative correlation with feeding damage by thrips; however, correlation does not mean causation. Thus, other associated characteristics may be involved in the mechanism of resistance. Meanwhile, papillae density may provide an easy marker in Gladiolus-breeding programs targeted at increased resistance to thrips.

## 4. Materials and Methods

### 4.1. Plant Materials

Fourteen different Gladiolus varieties differing in size were used. Six small varieties (Charming, Charming Beauty, Nymph, Alba, Elvira, and Robinetta) were obtained from Gebr P. & M. Hermans (Lisse, The Netherlands) and eight medium-to-large size varieties (Ben Venuto, Red Balance, V-29, Chinon, Live Oak, Deepest Red, Green Star, and Essential) were obtained from VWS B.V. (Alkmaar, The Netherlands). Each bulb was planted into a 9 × 9 cm^2^ pot filled with a 1:1 mixture of potting soil and dune sand. Six to ten replicates of each variety were randomly placed in a growth room (L:D, 18:6, 20 °C) and grown for 10 weeks. Three to five replicates of each variety were used for a whole-plant thrips bioassay, while the remaining replicates were used for measuring morphological parameters and metabolomic analysis.

### 4.2. Plant Resistance to Thrips

A non-choice whole-plant bioassay was conducted, as described in Leiss et al. [25]. Plants were placed individually in a thrips-proof cage, consisting of a plastic cylinder (80 cm height, 20 cm diameter), closed with a displaceable ring of thrips-proof gauze. The cages were arranged in a fully randomized design in a climate chamber (L18: D6, 20 °C). Next, 2 male and 18 female adult western flower thrips were added and left for 2 weeks. Thereafter, silver damage, expressed as the leaf area damaged in mm^2^, was visually scored for each plant.

### 4.3. Morphological Measurements

Morphological resistance traits were measured on the longest leaf of each replicate. We measured the length of the leaves, length of epidermal cells and mesophylls as well as the density of the epicuticular papillae, which form a convex outgrowth of the epidermal cells. The density of papillae was measured as the number of papillae per 2100 µm^2^. To measure these traits, cross sections of fresh leaves were examined under a confocal laser scanning and a visual light microscope (Zeiss LSM Exciter) with 20× magnification. Measurements were conducted using ImageJ software. To visualize the leaf surface of Gladiolus varieties, we selected Charming Beauty and Robinetta as representatives of a variety with high and low thrips damage, respectively, for scanning electron microscopy (SEM). We used a JSM6400 scanning electron microscope (JEOL; Tokyo, Japan). Leaf discs were fixed in 2.5% glutaraldehyde in 0.1 M phosphate buffer (pH 7), followed by dehydration with a graded series of acetone solutions (70%, 80%, 90%, 96%, and 100% acetone) for 10 min each. Before imaging, specimens were oriented, mounted on metal stubs, and sputter-coated with gold (Polaron 5000 sputtering system). In addition, the plant dry mass was measured after drying plants for 3 days in an oven at 50 °C.

### 4.4. In Vitro Thrips Bioassay

We first conducted an in vitro thrips bioassay using leaf extracts to investigate the potential effects of plant defense compounds. For this bioassay, we used 6 varieties that differed in susceptibility in the whole-plant bioassay described: Charming, Charming Beauty, Nymph, Alba, Elvira, and Robinetta. Briefly, 50 mg of dried leaf material of five replicates per variety was weighed and pooled for chemical extraction. The samples were extracted with 50% methanol in an ultrasonic bath for 20 min. After filtration, the residue was extracted again. The extraction was repeated three times, and the final filtrate was dried in a rotary evaporator. Next, 9 mg/mL of these extracts was re-dissolved in a 5% methanol–water solution, and the pH was adjusted to 7 [27].

For the in vitro thrips bioassay (Appendix A), 96-well plates were filled with 150 of µL 2% agarose and 50 µL of the extracts to be tested. Methanol—water (5%) and the insecticide abamectin (50 µg/mL) were used as a negative and a positive control, respectively. Each bioassay consisted of 32 replicates with 1 column of 8 wells on each of 4 plates. A single first instar thrips larva was placed into each cap of an 8-cap flat-cap strip. Each cap was sealed with parafilm through which the thrips could feed. The strips were then placed on top of the 96 -well plates. All the wells in one column of the 96-well plates received the same treatment. One variety, therefore, consisted of 8 replicates. Each variety was added to columns of 4 different 96-well plates, thus yielding 32 wells for each variety. An adhesive sealing film was placed onto the plates to prevent evaporation and to protect the samples during the assay. All plates were placed up-side down for 48 h to ensure that the thrips got into contact with the extracts. The plates were randomly placed in a growth chamber with standard thrips-rearing conditions (L18: D6, 23°C, 65% RH). After 48 h, the mortality of the thrips was recorded. Differences in thrips mortality among varieties were statistically analyzed with a chi-square test. The correlation between thrips mortality in the in vitro bioassay and thrips silver damage in the whole-plant non- choice bioassay was analyzed using Pearson correlation tests.

### 4.5. Metabolic Profiling

#### 4.5.1. Extraction of Plant Materials for NMR Metabolomics

Three replicates of leaves of each of the fourteen varieties were used for NMR metabolomics. The standard protocol of sample preparation and ^1^H NMR profiling described by Kim et al. [52] was applied. Samples of 30 mg of freeze-dried plant material were weighed into a 2 mL microtube and extracted with 1.5 mL of a mixture of phosphate buffer (pH 6.0) in deuterium oxide containing 0.05% trimethylsilylproprionic acid sodium salt-*d_4_* (TMSP) and methanol-*d_4_* (1:1). Samples were vortexed at room temperature for 1 min, ultrasonicated for 20 min, and centrifuged at 13,000 rpm for 10 min. An aliquot of 0.8 mL of the supernatant was transferred to 5 mm NMR tubes for ^1^H NMR measurement.

#### 4.5.2. NMR Analysis

^1^H NMR spectra were recorded with a 500 MHz Bruker DMX-500 spectrometer (Bruker, Karlsruhe, Germany) operating at a proton NMR frequency of 500.13 MHz. Deuterated methanol was used as the internal lock. Each ^1^H NMR spectrum consisted of 128 scans, requiring 10 min and 26 s acquisition time, with the following parameters: 0.16 Hz/point, pulse width (PW) of 30° (11.3 µs), and relaxation delay (RD) of 1.5 s. A pre-saturation sequence was used to suppress the residual water signal with low-power selective irradiation at the water frequency during the recycle delay. Free induction decay (FID) was Fourier-transformed with a line broadening (LB) of 0.3 Hz. The resulting spectra were manually phased and baseline-corrected to the internal standard TMSP at 0.00 ppm using TOPSPIN version 3.5 (Bruker). Two-dimensional J-resolved NMR spectra were acquired using 8 scans per 128 increments for F_1_ (chemical shift axis) and 8 k for F_2_ (spin–spin coupling constant axis) using spectral widths of 66 and 5000 Hz, respectively. Both dimensions were multiplied by sine-bell functions (SSB = 0) prior to double-complex Fourier transformation. J-resolved spectra were tilted by 45°, symmetrized about F_1_, and then calibrated to TMSP using XWIN NMR version 3.5 (Bruker). ^1^H–^1^H-correlated COSY spectra were acquired with a 1.0 s relaxation delay and a 6361 Hz spectral width in both dimensions. The window function for the COSY spectra was Qsine (SSB = 0).

#### 4.5.3. Data Processing

For the NMR spectra, intensities were scaled to total intensity and reduced to integrated equal widths (0.04 ppm) corresponding to the region of δ 0.32–10.0 (Appendix A). The regions of δ 4.7–5.0 and δ 3.30–3.34 were excluded from analysis due to the presence of the residual signals of water and methanol. ^1^H NMR spectra were automatically binned by AMIX software version 3.7 (Biospin, Bruker). Data were further analyzed with principal component analysis (PCA) and orthogonal partial least squares–discriminant analysis (OPLS-DA) using SIMCA-P software version 12.0 (Umetrics, Umea, Sweden). One of unsupervised multivariate methods is principal component analysis (PCA). It is used to reduce the dimensionality of a multivariate dataset. However, there is a limitation to see minor separation, because PCA could extract grouping information only from the maximum separation on the signals in the spectra representing the metabolomes. To solve the limitation of PCA, a supervised multivariate data analysis, orthogonal partial least squares (OPLS)—discriminant analysis (DA), in which another dataset of resistance was correlated with the chemical dataset, was performed. Pareto scaling was used for PCA and unit variance scaling for OPLS-DA. 

### 4.6. Statistical Analysis

Differences between varieties in morphological traits and plant dry mass were analyzed with one-way ANOVA and subsequent post hoc analysis with Bonferroni correction. Silver damage did not fit a normal distribution and was, therefore, ln-transformed. Correlations between thrips silver damage and morphological traits were analyzed using Pearson correlation.

Differences between Gladiolus varieties in the relative concentrations of metabolites related to thrips resistance were analyzed by one-way ANOVA. Pearson correlations were calculated for the relationships between metabolite concentrations, thrips silver damage, and density of epicuticular papillae. For epicatechin, epigallocatechin, and gallic acid, Kruskal–Wallis and Spearman rank correlations were used because data were not normally distributed.

## 5. Conclusions

Our study suggests that chemical compounds produced or stored in the epidermal papillae may confer thrips resistance to Gladiolus species. This offers an existing promise for further research on the mechanisms involved. Meanwhile, the density of papillae may provide an easy marker in Gladiolus-breeding programs targeted at increased resistance to thrips.

## Figures and Tables

**Figure 1 plants-10-01384-f001:**
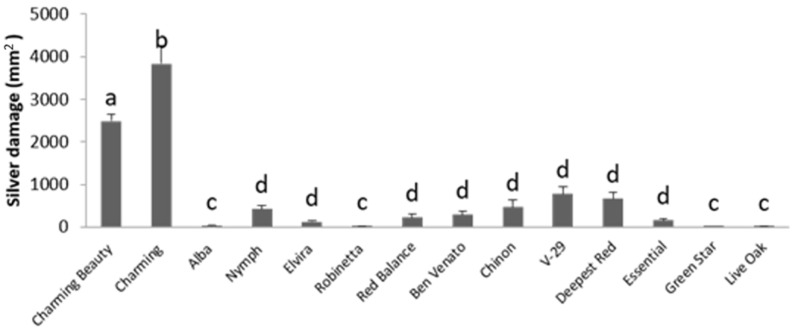
Silver damage (mm^2^) in 14 Gladiolus varieties, as measured by a whole-plant thrips non-choice bioassay. Data represent means and standard errors for three to five replicates. Different letters indicate significant differences between varieties at *p* ≤ 0.05.

**Figure 2 plants-10-01384-f002:**
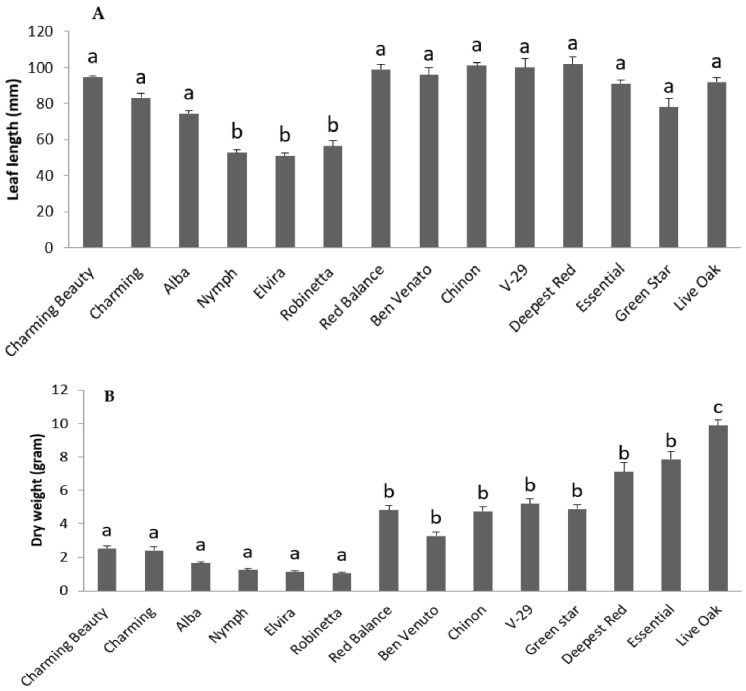
Leaf length (**A**) and dry mass (**B**) of 14 Gladiolus varieties. Data represent means and standard errors for three to five replicates. Different letters indicate significant differences between varieties at *p* ≤ 0.05.

**Figure 3 plants-10-01384-f003:**
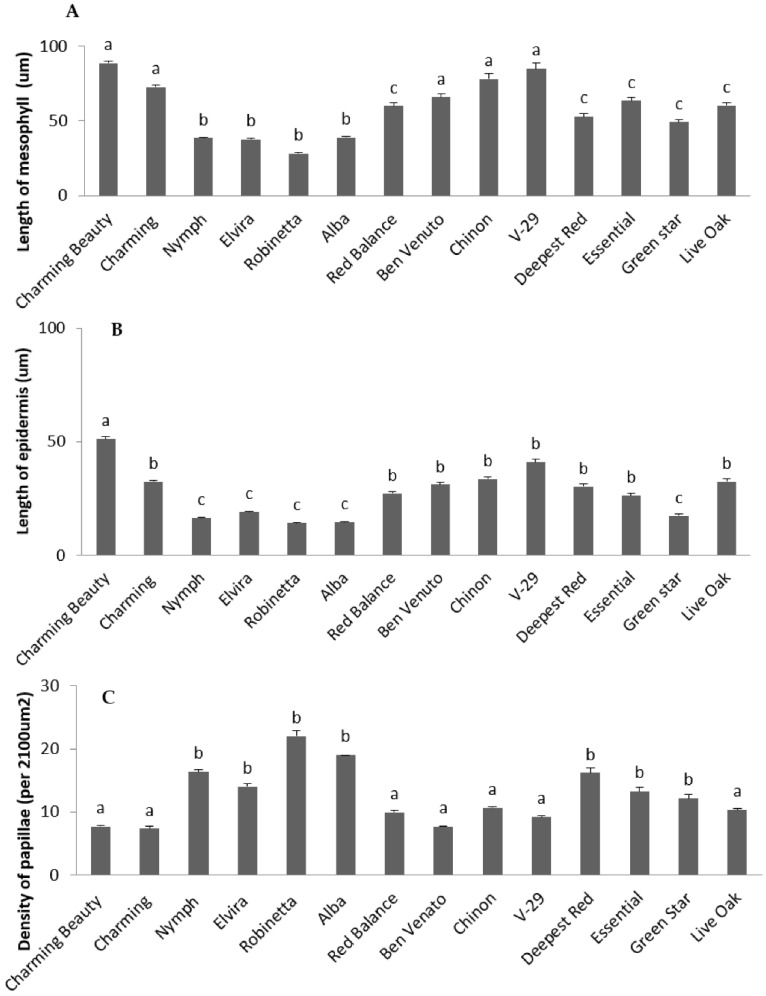
The length of mesophylls (**A**) and epidermal cells (**B**) and the density of papillae (**C**) in 14 Gladiolus varieties. Data represent means and standard errors for three to five replicates. Different letters indicate significant differences between varieties at *p* ≤ 0.05.

**Figure 4 plants-10-01384-f004:**
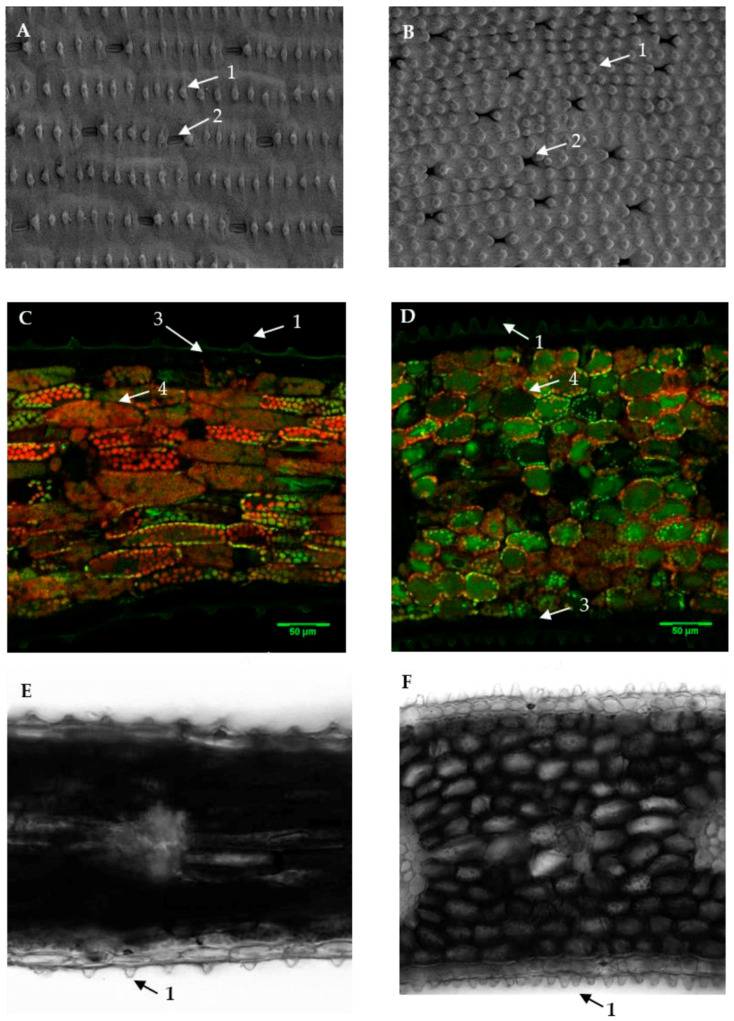
Leaf surface scanning electron photomicrographs of the thrips-susceptible Gladiolus variety Charming Beauty (**A**) and the thrips-resistant variety Robinetta (**B**). Lead cross sections of Charming Beauty (**C**) and Robinetta (**D**) with confocal laser scanning. Leaf cross sections of Charming Beauty (**E**) and Robinetta (**F**) with visual light microscopy. Arrows indicate papillae (1), stomata (2), epidermal cells, (3) and mesophyll (4).

**Figure 5 plants-10-01384-f005:**
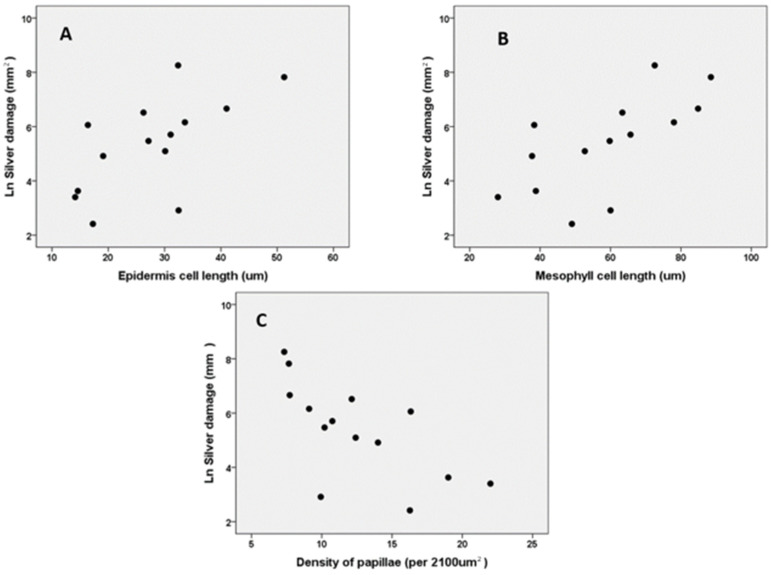
Correlation between thrips silver damage, measured in a whole-plant non-choice bioassay, and cell length of epidermal cells (**A**) (*r* = 0.596, *N* = 14, *p* = 0.024) and mesophylls (**B**) (*r* = 0.603, *N* = 14, *p* = 0.022) and density of epidermal papillae (**C**) (*r* = −0.628, *N* = 14, *p* = 0.016) in 14 Gladiolus varieties.

**Figure 6 plants-10-01384-f006:**
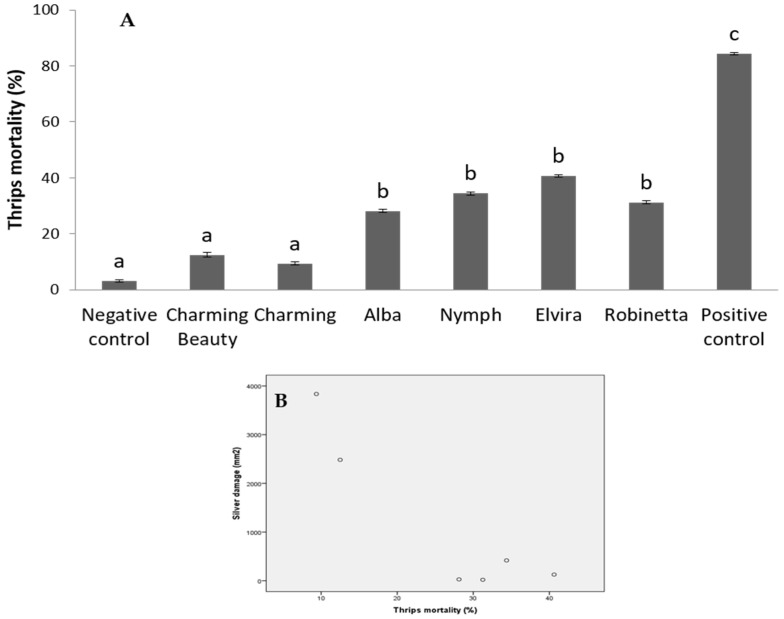
(**A**) Mortality of thrips feeding on artificial diets (150 µL of 2% agarose) with 50 µL of leaf extracts of six *Gladiolus* varieties measured in an in vitro bioassay. For each extract, 32 thrips were tested 5% methanol solution was used as a negative control and the insecticide abamectin (50 µg/mL) as a positive control. Means and standard errors are presented. Different letters indicate significant differences between varieties at *p* ≤ 0.05. (**B**) Correlation between thrips silver damage, measured in a whole-plant non-choice bioassay, and thrips mortality measured in the in vitro bioassay of six *Gladiolus* varieties (*r* = −0.788, *N* = 6, *p* = 0.031).

**Figure 7 plants-10-01384-f007:**
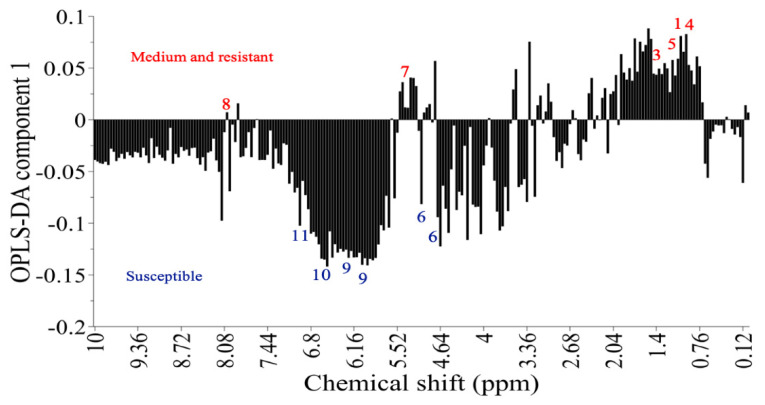
Loading plot for OPLS-DA of Gladiolus varieties based on ^1^H NMR spectra. Metabolites are labeled as (1) signal A, (2) signal B, (3) alanine, (4) valine, (5) threonine, (6) glucose, (7) sucrose, (8) kaempferol, (9) epicatechin, (10) epigallocatechin, and (11) gallic acid.

**Table 1 plants-10-01384-t001:** Pearson correlations (*N* = 14) between ln-thrips silver damage (mm^2^) and epidermal cell length (µm), mesophyll length (µm), density of papillae (per 2100 µm^2^), leaf length (cm), and dry mass (g) in Gladiolus varieties (*N* = 14). Data represent means of three to five replicates. Data in bold show the significance level at *p* ≤ 0.05.

	Epidermal Cell Length (µm)	Mesophyll Length (µm)	Density of Papillae (per 2100 µm^2^)	Leaf Length (cm)	Dry Mass (g)
Ln silver damage	*r* = 0.596 *p* = 0.024	*r* = 0.603 *p* = 0.022	*r* = −0.628 *p* = 0.016	*r* = 0.320 *p* = 0.264	*r* = −0.222 *p* = 0.445
Epidermal cell length		*r* = 0.931 *p* = 0.000	*r* = −0.873 *p* = 0.000	*r* = 0.704 *p* = 0.005	*r* = 0.310 *p* = 0.281
Mesophyll length			*r* = −0.909 *p* = 0.000	*r* = 0.777 *p* = 0.001	*r* = 0.315 *p* = 0.273
Density of papillae				*r* = −0.669 *p* = 0.009	*r* = −0.389 *p* = 0.170
Leaf length					*r* = 0.441 *p* = 0.114

**Table 2 plants-10-01384-t002:** Correlations between the concentrations of metabolites that are not related to thrips resistance and silver damage and the density of papillae in 14 Gladiolus varieties. Silver damage was ln-transformed to obtain normally distributed data. * = *p* < 0.05 and *N* = 14 in all cases. Spearman correlations were used for the relationships between silver damage, epigallocatechin (EGC), gallic acid, and density of papillae.

	Valine	Sucrose	Glucose	EGC	Gallic Acid
Ln damage	*r* = −0.100 *p* = 0.734	*r* = 0.083 *p* = 0.779	*r* = 0.265 *p* = 0.360	*r* = 0.404 *p* = 0.152	*r* = 0.313 *p* = 0.276
Papillae	*r* = −0.034 *p* = 0.907	*r* = −0.117 *p* = 0.692	*r* = −0.442 *p* = 0.114	*r* = −0.275 *p* = 0.342	*r* = −0.019 *p* = 0.950
Valine		*r* = 0.345 *p* = 0.227	*r* = 0.096 *p* = 0. 743	*r* = −0.594 * *p* = 0. 025	*r* = −0.403 *p* = 0.153
Sucrose			*r* = −0.123 *p* = 0.676	*r* = −0.576 * *p* = 0.031	*r* = −0.058 *p* = 0.845
Glucose				*r* = −0.047 *p* = 0.874	*r* = 0.139 *p* = 0.636
EGC					*r* = 0.074 *p* = 0.801

## Data Availability

^1^H NMR data of the samples are deposited in the data storage of Natural Products Laboratory (Institute of Biology, Leiden University, Leiden, The Netherlands). The data can be provided upon request.

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
