# Peer review of "Morphological and Chemical Factors Related to Western Flower Thrips Resistance in the Ornamental Gladiolus"

_plants, 2021, doi:10.3390/plants10071384_

Round 1

Reviewer 1 Report

The authors (Wahyuni et al.) are interested in identifying gladiola traits (leaf morphological and chemical) that could be used to reliably evaluate gladiola varietal resistance/susceptibility to western flower thrips (Frankliniella occidentalis). They examined leaf cell types, used NMR to broadly identify groups of chemicals (amino acids, terpinoid saponins, sucrose...), and performed thrips feeding assays (leaf extract toxicity assays) as a means to identify associations (statistical correlations) between thrips feeding damage (silver feeding scars), density of epidermal cell types, and concentration of leaf-extracted chemicals (NMR analysis and toxicity tests).   The authors identified morphological leaf traits (papillae density) to be significantly correlated with feeding damage (resistance to WFT), and found that particular chemistries correlated with morphological traits (cell types, papillae density), and that total leaf extracts affected larval thrips survival (% mortality in in vitro artificial feeding chambers).  A few components (two AA's and one unidentified component) generated from the NMR analysis were correlated with feeding damage by adults. The authors claimed that in a general sense, chemicals exuded from papillae (presuming extracted chemicals are only coming from papillae) are some how responsible for resistance to WFT. This reviewer does not feel this statement is justifiable given their results. First, total chemical extracts from freeze-dried leaves does not point directly to papillae-exuded chemistries. Many secondary compounds are produced in other leaf cell types, and disrupting tissue membranes by freeze-drying/pulverizing could induce chemical responses (wound-like responses) that may compound the toxic effect. Second, it is well described that WFT feeding induces host responses (gene expression and subsequent downstream defense compounds), and this may also contribute to resistance, and these types of induced innate defense responses were not considered in this study.  It is possible that other chemicals not accounted for by the NMR method are related to resistance to WFT (e.g., constitutively-expressed chemistries, e.g., acyl sugars, fatty acids, or feeding-inducible secondary metabolites). The authors do provide enough convincing evidence that papillae density is correlated with resistance (feeding damage).

This reviewer strongly suggests that the authors tone down their conclusion that chemical properties of the papillae indicated resistance to WFT (lines 497 and 665) and acknowledge alternative hypotheses in their discussion. This reviewer also commends the authors for including negative results instead of hand-picking the 'best' results to share.

Other major comments:
1. Missing literature in discussion related to other resistance mechanisms associated with leaf chemistry and WFT specifically - namely acyl sugars produced by trichomes and their relation to WFT performance and fitness, listed below; please include these because they are directly related to chemical basis of trichome-based (leaf and flower) host resistance to WFT. Also, the authors devoted a lot of discussion of their findings in the context of other herbivorous/pestivorous arthropods - is there no literature associated with WFT and host resistance or defense responses of relevance to this study? Plant monoterpines and thrips?

"A thrips vector of tomato spotted wilt virus responds to tomato acylsugar chemical diversity with reduced oviposition and virus inoculation." November 2019, Scientific Reports 9(1):17157. DOI: 10.1038/s41598-019-53473-y

"Acylsugar amount and fatty acid profile differentially suppress oviposition by western flower thrips, Frankliniella occidentalis, on tomato and interspecific hybrid flowers."
July 2018, PLoS ONE 13(7):e0201583. DOI: 10.1371/journal.pone.0201583

2. Missing background information of WFT feeding and damage caused by thrips in the Introduction. For readers unfamiliar with WFT or thrips in general, this background information would help readers understand the rationale of the measurements used. Please explicitly define resistance to thrips - the authors seem to use the word 'resistance' to mean both leaf traits and extent of feeding damage and it's not clear which they intend to infer in the text. Also, is thrips feeding damage the best and or only measure of thrips performance and measure of host resistance? Do both males and females produce silver feeding scars?

3. In vitro, chemical extract toxicity test (lines 599-612)- needs a more clear description of the methodology, feeding chamber set up, experimental design - how many varieties represented in one 8-well strip? If only one, then these 8 are not randomly assigned (thus, one replicate strip with 8 technical reps/strip). How many strips per variety? Please explain and clarify the experimental design behind the statistical analysis. This methods sections needs improvement to clearly understand the assay parameters. This reviewer suggests a supplemental figure of the feed assay/toxicity set-up. Also, why first instar larval thrips in feeding assay and adults in whole plant assay? How do the authors justify correlating L1 survivorship in the artificial diet assay and feeding damage caused by adults in the whole plant assay (Fig. 6)? This reviewer is not convinced that a correlation analysis of these two responses are appropriate (lines 610-612; line 247). 

4. Multivariate analysis tools require clear descriptions of their utility in the Methods sections (lines 646 - 651). PCA is commonly used and is a non-biased, soft-clustering method for reducing multiple dimensions (loading variables) into a few explanatory components (PCs) that explain the majority of the variation in the multivariate data set, and then can be used to visualize the separation of treatments. Please include the PCA plots, PC scores and loadings (correlations between original variable response - NMR spectral intensities? and PC scores) as supplementary data since the authors mention that the analysis did not differentiate differences among resistance phenotypes, and would like to evaluate the outcome of this analysis. What were the loading variables (peak intensities?) and how did the authors handle the treatments? Were the treatments just the varieties, and loading variables the NMR peak intensities? Or chemical groups identified? It is not clear. And what does PLS-DA and S-plot offer? Many readers will not understand the significance and utility of these exploratory methods. Please explain.

5. Figure 7 legend needs improvement. What are the axis abbreviations? What does this plot tell us, like what are the red dots vs black dots? Again, a description of these in the legend AND a more complete description of this tool and how it is interpreted should be added in Methods section since it was used to drill down of a key components of the leaf extract for correlation analyses.

6. Figure 4 E - H panels of leaf cross sections are too dark and of poor quality. One can achieve better resolution with thinner sections and it is very difficult to discern the importance (arrows) of what the author intends to show here. Please improve the quality of the cross-section (acquire new sections) and microscopic images if these are essential for evaluating/understanding the data and interpretation of this type of data.

7. Signals related to resistance and susceptibility (lines 256-390)- lots of redundancy in text and this reviewer suggests that authors report on what IS significantly correlated with damage and morphological traits, and let the reader use table 2 to discern what is NOT correlated. The writing recapitulates the results in table 2 unnecessarily. Please revise this section in the results to improve clarity and flow.

Author Response

Dear Reviewer,

Hereby a revised manuscript based on the comments. Please see the attachment.

Kind Regards,

Dinar

Reviewer 2 Report

In this manuscript the authors investigated the resistance mechanisms of Gladiolus species against Western flower thrips. Studying both the morphological traits of various plants and the secondary metabolites they produced in correlation with their resistance. Although the morphological analysis, and the conclusions drawn are seems to be justified, a few questions arisen regarding the NMR measurements of the extracts:

  • In the materials and methods part, the authors state, that they prepared the NMR samples from "freeze-dried plant material". Could you specify which parts of the plants were used?
  • I miss the 1H-NMR spectra measured of the 14 variants. It should be included in the supplementary material.
  • I assume, that the plant extract was a complex mixture of various compounds, providing a complicated NMR spectrum. I do not find it reassuring, that alanine and threonine were identified by a simple signal each in the aliphatic region of the spectrum. At least reference spectra is required in both cases in the same NMR solvent mixture. What about the other signals of these compounds, did the authors find them in the spectrum? For example, the hydrogen atoms on the alpha carbons should appear around 3 or 4 ppm.
  • Without the NMR spectra, I am not convinced, that there were no overlaps between charactheristic signals used for the measurement of relative concentrations

Overall, I find this work suitable for publication in Plants, after discussing the above mentioned issues and the replacement of the missing spectra. 

Author Response

Dear Reviewer,

Here is a revised manuscript, Please find it attached.

Round 2

Reviewer 2 Report

Thank you for your replies, I accept the answers. I agree, that you should provide the original bucket files of NMR data as a separate supplementary file, so the readers may have access to raw data and have their own observations and conclusions. 

Author Response

Dear Academic Editor,

Please find the second-round comments in the attachment. 
